# Topsoil Nutrients Drive Leaf Carbon and Nitrogen Concentrations of a Desert Phreatophyte in Habitats with Different Shallow Groundwater Depths

**Bo Zhang** [1,2,3,*] , **Gangliang Tang** [1,3,4] , **Hanlin Luo** [1,3,5] , **Hui Yin** [1,3,4] , **Zhihao Zhang** [1,3,4] , **Jie Xue** [1,3,4] , **Caibian Huang** [1,3,4] , **Yan Lu** [1,3,4] , **Muhammad Shareef** [1,3,4] , **Xiaopeng Gao** [6] **and Fanjiang Zeng** [1,3,4,*]

[1] Xinjiang Key Laboratory of Desert Plant Roots Ecology and Vegetation Restoration, Xinjiang Institute of Ecology and Geography, Chinese Academy of Sciences, Urumqi 830011, China; tanggangliang@ms.xjb.ac.cn (G.T.); huangmozhixing@163.com (H.L.); yinhui@ms.xjb.ac.cn (H.Y.); zhangzh@ms.xjb.ac.cn (Z.Z.); xuejie11@ms.xjb.ac.cn (J.X.); huangcaibian@ms.xjb.ac.cn (C.H.); luyan@ms.xjb.ac.cn (Y.L.); shareefagron@gmail.com (M.S.)
[2] National Engineering Technology Research Center for Desert-Oasis Ecological Construction, Xinjiang Institute of Ecology and Geography, Chinese Academy of Sciences, Urumqi 830011, China
[3] Cele National Station of Observation and Research for Desert-Grassland Ecosystems, Cele 848300, China
[4] State Key Laboratory of Desert and Oasis Ecology, Xinjiang Institute of Ecology and Geography, Chinese Academy of Sciences, Urumqi 830011, China
[5] Lanzhou New District Urban Development Investment Group Co., Ltd., Lanzhou 730000, China
[6] Department of Soil Science, University of Manitoba, Winnipeg, MB R3T 2N2, Canada; gaoxp@umanitoba.ca
[*] Correspondence: zhangbo@ms.xjb.ac.cn (B.Z.); zengfj@ms.xjb.ac.cn (F.Z.)

**Abstract:** Phreatophytes are deep-rooted plants that reach groundwater and are widely distributed in arid and semiarid areas around the world. Multiple environmental factors affect the growth of phreatophytes in desert ecosystems. However, the key factor determining the leaf nutrients of phreatophytes in arid regions remains elusive. This study aimed to reveal the key factors affecting the ecological stoichiometry of desert phreatophytes in the shallow groundwater of three oases at the southern rim of the Taklimakan Desert in Central Asia. Groundwater depth; groundwater pH and the degree of mineralization of groundwater; topsoil pH and salt concentration; topsoil and leaf carbon, nitrogen, and phosphorus concentrations of phreatophytic *Alhagi sparsifolia* grown at groundwater depths of 1.3–2.2 m in the saturated aquifer zone in a desert–oasis ecotone in northwestern China were investigated. Groundwater depth was closely related to the mineralization degree of groundwater, topsoil C and P concentrations, and topsoil salt content and pH. The ecological stoichiometry of *A. sparsifolia* was influenced by depth, pH and the degree of mineralization of groundwater, soil nutrients and salt concentration. However, the effects of soil C and P concentrations on the leaf C and N concentrations of *A. sparsifolia* were higher than those of groundwater depth and pH and soil salt concentration. Moreover, *A. sparsifolia* absorbed more N in the soil than in the groundwater and atmosphere. This quantitative study provides new insights into the nutrient utilization of a desert phreatophyte grown at shallow groundwater depths in extremely arid desert ecosystems.

**Keywords:** ecological stoichiometry; *Alhagi sparsifolia*; groundwater table; soil salt; extremely arid region; ecological protection

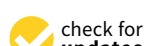



## 1. Introduction

Carbon, nitrogen, and phosphorus are vital microelements in most terrestrial plants. C, which comes from $CO_2$ in the air by plant photosynthesis, is the foundation for synthesizing nearly all organic compounds. Nitrogen is a critical component of proteins, nucleic acids, nucleotides, and enzymes as well as amino acid synthesis. Phosphorus is a vital element of phospholipids, water-soluble P esters, phosphoproteins, DNA, and RNA [1,2]. C, N, and P concentrations in plants are strongly related to plant growth status [1], and the N:P ratio in

plants is widely used to indicate nutrient limitation in various ecosystems [3,4]. In addition, plant elemental content and ratio variations are closely related to ecosystem processes such as productivity, energy flow, plant growth strategy, and decomposition [5,6]. Therefore, plant elemental content and their ratio variations are useful tools for understanding the mechanisms of plant growth status and ecosystem functions in different ecosystems.

Plant roots that access groundwater to obtain water can be considered phreatophytes. Phreatophytes are a hydroecological plant type that is widely distributed in arid and semiarid areas in grasslands and deserts around the world, except Antarctica [7]. Through their deep roots, phreatophytes have remarkable advantages in regions with insufficient precipitation. Therefore, phreatophytes can adapt better to arid environments than other plants [5,8,9]. *Alhagi sparsifolia* Shap., a dominant perennial phreatophyte grown at the southern rim of the Taklimakan Desert, has an important influence on wind prevention and sand fixation in the local ecosystem. *A. sparsifolia* is a leguminous plant and is an economically important species for local livestock because of its high protein concentration [5,8,10,11]. The deep roots of *A. sparsifolia* usually reach 20 m to meet groundwater, which is the main source of water and nutrients [12,13].

Numerous studies have identified several factors affecting plant growth and nutrient concentrations in *A. sparsifolia* [5,8,10,14–21]. First, previous studies have shown that groundwater is a major source of nutrients and water for *A. sparsifolia*. The xylem sap concentrations of $NO_3$ and $PO_3$ in *A. sparsifolia* indicate a sufficient nutrient supply via groundwater [7,12,13]. Therefore, the variation in groundwater depth has a significant effect on the biomass, cover, growth rate, leaf P concentration, and nutrient resorption of *A. sparsifolia* [5,9,15,16]. Second, high groundwater salinity remarkable alters biomass and species diversity [19,22,23]. The biomass of *A. sparsifolia* declines with increasing groundwater salinity on shallow groundwater [23], and low groundwater salinity is closely related to high plant diversity [19]. Third, soil nutrients influence plant growth and leaf nutrients of *A. sparsifolia* in different natural habitats [5,8,10,17]. The leaf N and P concentrations of *A. sparsifolia* are positively related to soil N and P concentrations [5,9,10,17]. Soil nutrients are the main driving factors affecting the bacteria and fungus communities of *A. sparsifolia* in arid desert ecosystems [20]. Finally, soil salinity notably affects the population density and nitrate assimilation of *A. sparsifolia*. The population density of *A. sparsifolia* is approximately 20% at a slight soil salinity and drops to less than 15% under severe soil salinity [14]. The mineral nitrogen assimilation of *A. sparsifolia* is more sensitive to soil salinity than biological N fixation [21]. *A. sparsifolia* rehabilitates saline soil environments because of its high ion selectivity, which excludes $Na^+$ and accumulates $Ca^{2+}$ in its leaves [18]. However, the key factor that affects leaf C, N, and P concentrations in *A. sparsifolia* is still unknown.

The Taklimakan Desert, which covers 37,000 square kilometers, is the world's second-largest shifting sand desert and China's largest desert. Previous reports have shown that phreatophytic desert plants always reach the groundwater to obtain water and nutrient resources and make them dependent on water and nutrients from soil in the Taklimakan Desert [7,12,13,21]; however, studies revealing the most important factor that determines the leaf nutrients of phreatophytes are relatively few. In this study, we chose five study sites in three desert–oasis ecotones at the southern rim of the Taklimakan Desert to determine the environmental factors, such as groundwater depth and salinity, soil nutrients, and salinity, that play a crucial role in influencing the leaf C, N, and P concentrations of *A. sparsifolia* in an arid ecosystem. Our hypothesis was that groundwater depth has the most important effect on the variation of nutrients in *A. sparsifolia* in various shallow groundwater habitats.

## 2. Materials and Methods

### 2.1. Study Area

The study sites were located in Cele, Moyu, and Hetian counties in the Hetian Prefecture (79°28′ E–81°04′ E, 36°54′ N–37°14′ N), which belong to the southern rim of the Taklimakan Desert of the Xinjiang Uyghur Autonomous Region, China (Figure S1). The Taklimakan Desert is located in the Tarim Basin in western China and in the central part of

Asia within the rain shadows of the surrounding Tian Shan and Kunlun mountain ranges. Consequently, the climate is extremely arid in this area. The mean annual precipitation ranges from 20 to 70 mm, and the annual potential evapotranspiration is approximately 2600 mm. The highest and the lowest temperatures are 41.9 °C in July and −23.9 °C in January, respectively [9]. Vegetation, including the herbaceous legume *Alhagi sparsifolia* Shap. (Fabaceae), herbaceous perennial *Karelinia caspia* (Pall.) Less. (Asteraceae), salt cedar *Tamarix ramosissima* Ledeb. (Tamaricaceae), *Phragmites australis* (Cav.) *Trin. ex Steu.* (Poaceae), and poplar tree *Populus euphratica* Oliv. (Salicaceae), is sparse in the five research sites. The groundwater depth ranges from 1.3 to 2.2 m in the saturated aquifer zone. The soils are aeolian sandy soil in these five research sites. The soil is Aridisol in the USDA ST system and contains 90% sand, 4% clay, and 6% silt at a 0–20 cm soil depth [24].

### 2.2. Methods

Each site had four quadrats that covered $10 \times 10$ m. During the experimental period from May to August 2016, four samples of soil (20 cm depth), groundwater, and plant samples were collected monthly. The samples were dried to a constant weight and maintained at 70 °C. After natural air drying, soil samples were treated using the quartering method and a 2 mm soil sieve. Soil water content was determined using a drying method. Soil salinity was measured using the residue-drying method. Air-dried soil samples were passed through a 2 mm sieve, and a 10.0 g soil sample was added to 50 mL ultrapure water (soil: water ratio = 1:5). The mixture was shaken for 3 min. Then, 20 mL of the supernatant was placed in a weighed beaker and dried [25]. Soil pH was assayed using the Mettler Toledo S20K pH meter (Mettler Toledo Instruments, Greisensee, Switzerland). Soil organic carbon was determined using the $K_2Cr_2O_7$ digestion method. Total N and P contents were determined using the Kjeldahl and digestion methods, respectively [8]. Groundwater pH was assayed using a Mettler Toledo S20K pH meter (Mettler Toledo Instruments, Greisensee, Switzerland). The degree of mineralization of the groundwater was determined using a drying method. Groundwater (100 mL) was placed in a small beaker and dried in a steam bath. If a colored steaming residue was observed, the sample was added with a few drops of (1 + 1) hydrogen peroxide solution, dried in a steam bath until the color of the residue did not change, baked at 105 °C for 2 h, and weighed [26]. The leaf organic content was determined using a total organic carbon analyzer (Aurora 1030; College Station, TX, USA). Leaf N and P contents were determined using the Kjeldahl and digestion methods, respectively [8].

### 2.3. Data Analysis

One-way ANOVA with Tukey's test was carried out for the multiple comparisons of groundwater depth, groundwater pH, and the degree of mineralization of groundwater; soil salt, soil pH, and C, N, and P concentrations of soil and *A. sparsifolia* at different research sites to test statistical significance ($p < 0.05$). Linear relationships among variables (i.e., soil, plant, and groundwater) were analyzed using the "corrplot" package in R software (version 4.0.3). The statistical significance values of each explanatory variable, including groundwater and soil variables, were tested using the "rdacca.hp" package in the R software [27]. The structural equation model (SEM) was used to tease apart statistically and quantify the direct and indirect effects of groundwater and soil variables on the leaf nutrient concentration using the R software with the package "piecewiseSEM" [9,28]. The D-separation test of piecewise SEM was used to test whether the causal model missed important links, and a $p > 0.05$ indicated that the model was acceptable [29].

## 3. Results

### 3.1. Groundwater and Soil Characteristics

The mean and standard deviation for depth, pH, and the degree of mineralization of groundwater as well as soil pH, salt, C, N, and P for the different sites are shown in Figure 1. Significant variations in groundwater and soil characteristics were observed at

different study sites (Figure 1). The depth and pH of groundwater ranged from 1.30 to 2.20 m (Figure 1A) and 7.03 to 8.84 (Figure 1B), respectively. Groundwater depth was the highest in Hetian county (HT) and the lowest in northern Cele County (CN) among all study sites. The degree of mineralization of groundwater was the highest in CN and the lowest in the west of Cele County (CW) and HT among the five study sites. No significant differences were observed in the pH of the groundwater in this study. In addition, the soil pH was highest in HT and lowest in CW, Moyu County (MY), and CN among the five study sites. The soil salt concentration in CN was higher than that in the other study sites. Soil C concentrations were the highest in CN and the east of Cele County (CE) and the lowest in CW and HT, soil N concentrations were the highest in CN and the lowest in HT, and soil P concentrations were the highest in HT and the lowest in CW.

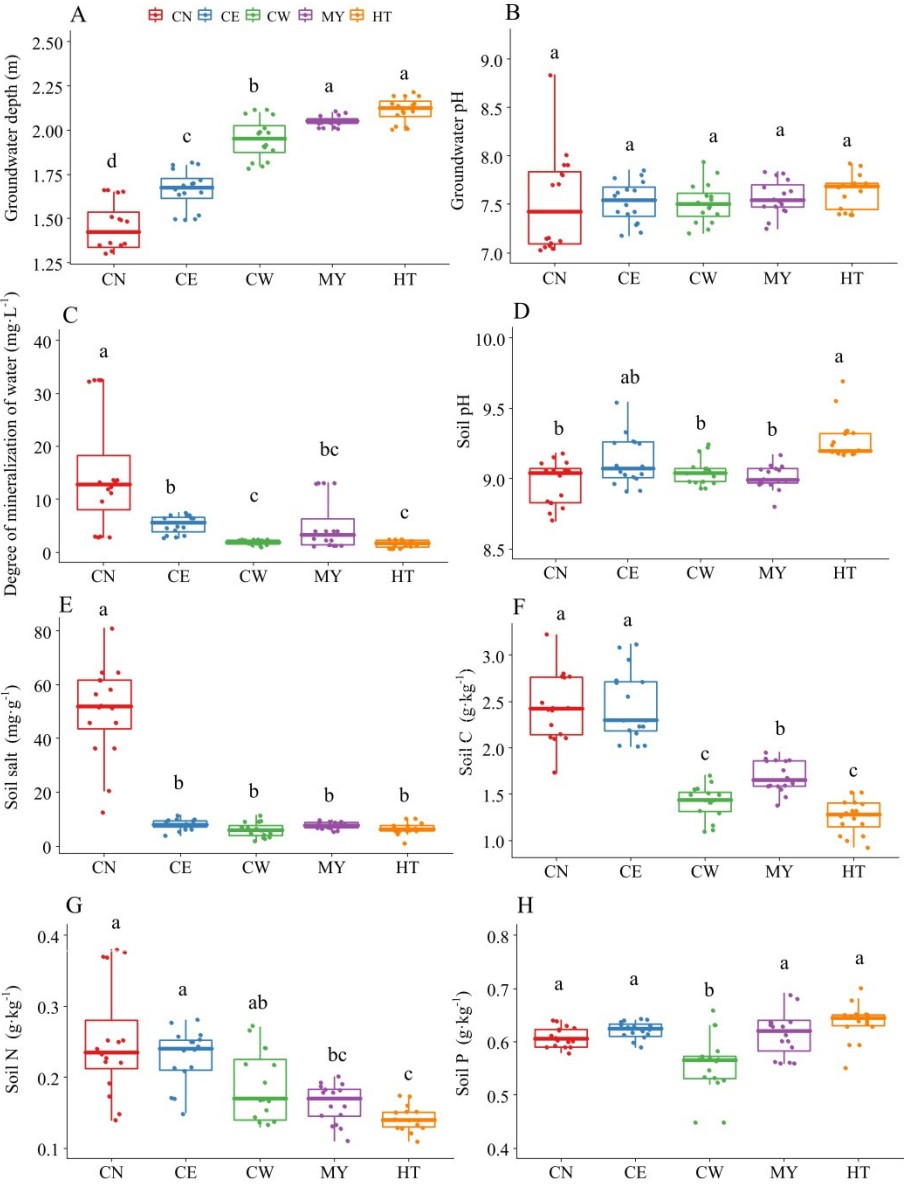

**Figure 1.** Changes in the environmental variables of the study sites. Vertical bars represent standard deviations, and the letters denote significant differences among different groundwater depths ($p < 0.05$). CW, west of Cele county; MY, Moyu county; HT, Hetian county; CN, north of Cele county; CE, east of Cele county. (**A–H**) represent variations of depth, pH, and the mineralization degree of groundwater as well as pH, salt, C, N, and P concentrations of soil.

### 3.2. Variations in Leaf Ecological Stoichiometry

The mean and standard deviation for leaf C, N, P, and the N:P ratio of *A. sparsifolia* for the different sites are shown in Figure 2. Significant differences in the leaf ecological stoichiometry of *A. sparsifolia* were observed at the study sites (Figure 2). In all study sites, leaf C concentrations were the highest in CE and the lowest in CW and HT; leaf N concentrations were the highest in CW and the lowest in HT; leaf N:P ratios were the highest in CW and the lowest in HT, CN, and CE. No significant differences were observed in the leaf P concentrations.

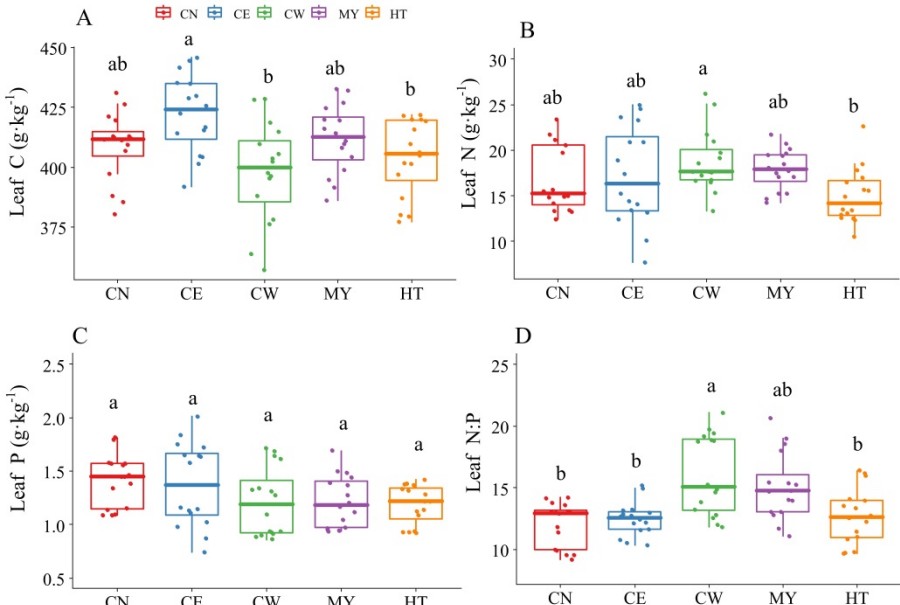

**Figure 2.** Changes in the leaf ecological stoichiometry of *A. sparsifolia* at the study sites. Vertical bars represent standard deviations, and the letters denote significant differences among different groundwater depths ($p < 0.05$). CW, west of Cele county; MY, Moyu county; HT, Hetian county; CN, north of Cele county; CE, east of Cele county. (**A–D**) represent variations of leaf C, N, P, and N:P ratio at the study sites.

### 3.3. Relationships among Groundwater, Soil, and Leaf Ecological Stoichiometry

Significant relationships ($p < 0.05$) were observed among groundwater depth, groundwater salinity, soil nutrient concentration, and soil salt concentration as well as leaf ecological stoichiometry of *A. sparsifolia* (Figure 3). Groundwater depth was significantly related with leaf P concentration and leaf N:P ratio. In addition, groundwater depth was significantly related to the degree of mineralization of groundwater, soil pH, soil salt, and soil C and N concentrations. A significant positive correlation was observed between the groundwater pH and leaf P concentration. Soil C concentration was strongly correlated to leaf P concentration and leaf N:P ratio. The soil N concentration was closely linked to leaf N:P ratio. The soil P concentration was negatively related to leaf N concentration. Soil salt concentration had a significant impact on leaf N:P ratio.

### 3.4. Relative Influences of Environmental Factors on Variation in Leaf Ecological Stoichiometry

The leaf C concentration of *A. sparsifolia* was largely determined by groundwater pH and soil P concentration (Figure 4A). Groundwater pH accounted for 39.9% of the variation in leaf C concentration followed by 18.9% variation in soil P concentration. The leaf P concentration was significantly influenced by groundwater table depth and soil C concentration, which explained 40.3% and 35.0% of variations in leaf P concentration, respectively (Figure 4C). The leaf ecological stoichiometry was strongly influenced by groundwater pH and soil P concentration, which contributed to 38.9% and 18.8% of

the variations, respectively, of leaf ecological stoichiometry (Figure 4D). No significant environmental factors affected leaf N concentrations (Figure 4B).

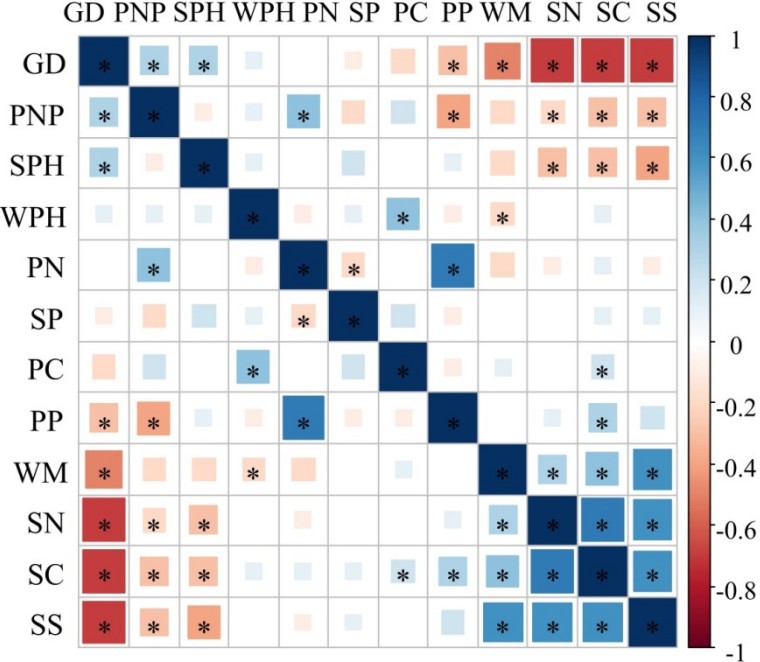

**Figure 3.** Relationships between groundwater, soil, and leaf ecological stoichiometry of *A. sparsifolia* at the study sites. GD, groundwater depth; WPH, groundwater pH; WM, degree of mineralization of groundwater; SS, soil salt; SPH, soil pH; SC, soil C concentration; SN, soil N concentration; SP, soil P concentration; PC, plant C concentration; PN, plant N concentration; PP, plant P concentration; PNP, plant N:P ratio. * Indicates significant correlation at $p < 0.05$.

### 3.5. Environmental Factors Affecting Variations in Leaf Ecological Stoichiometry

The key factors affecting the variations in leaf C, N, and P concentrations of *A. sparsifolia* were explored using SEM (Figure 5). The model explained 29%, 52%, and 15% of the variance in leaf C, N, and P concentrations, respectively. Groundwater depth had a direct effect (0.20) on leaf N concentration. Groundwater pH had direct effects (0.34) on the leaf C concentration and indirect effects (0.07) on the leaf N concentration. The soil C concentration had direct effects (0.45) on leaf C concentration and indirect effects (0.01) on leaf N concentration. In addition, the soil P concentration had a direct effect (0.21) on leaf N concentration. Therefore, the effects of soil nutrients (0.67) on variations in leaf C and N concentrations were higher than those of groundwater characteristics (0.61).

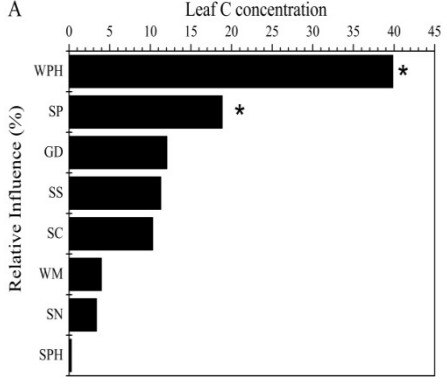
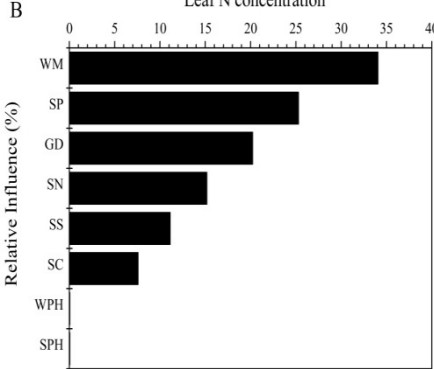

**Figure 4.** *Cont.*

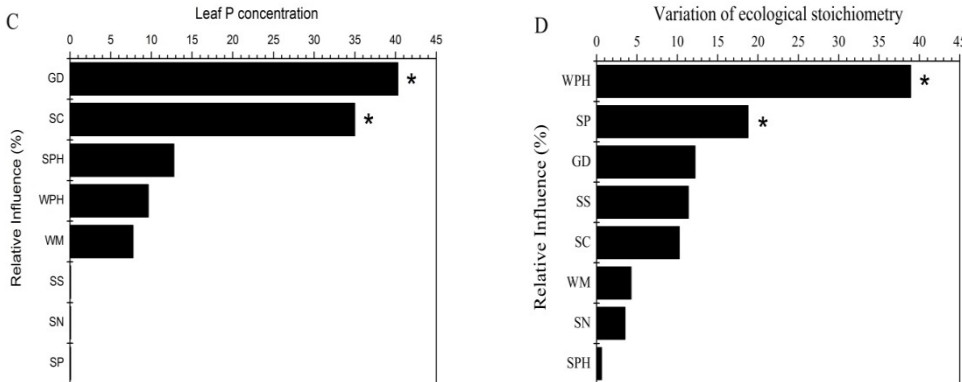

**Figure 4.** Relative influences of environmental factors on variations in leaf C, N, and P concentrations as well as ecological stoichiometry of *A. sparsifolia* in the study sites. GD, groundwater depth; WPH, groundwater pH; WM, the degree of mineralization of groundwater; SS, soil salt; SPH, soil pH; SC, soil C concentration; SN, soil N concentration; SP, soil P concentration; PC, plant C concentration; PN, plant N concentration; PP, plant P concentration; PNP, plant N:P ratio. * Indicates significant correlation at $p < 0.05$. (**A–D**) represent relative influences of environmental factors on variations in leaf C. N, P, and ecological stoichiometry of *A. sparsifolia*.

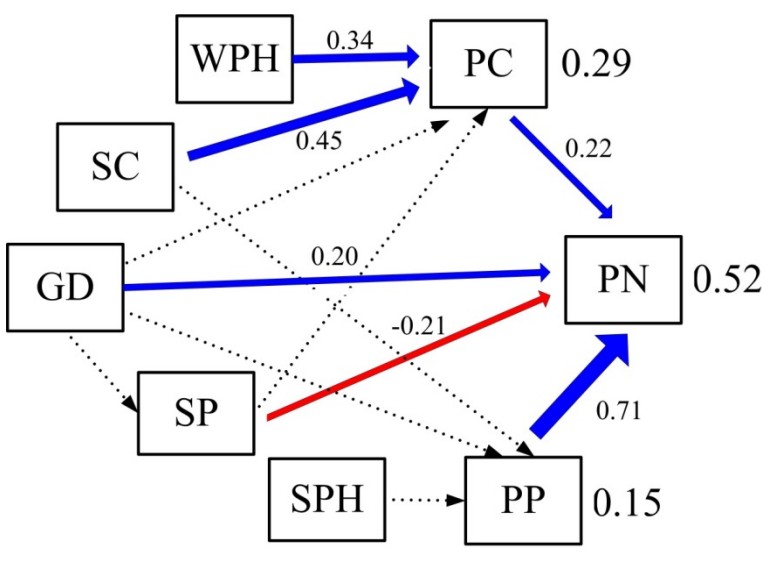

Fisher's C=27.708, *p*=0.067

**Figure 5.** Controlling factors analysis of variations in leaf C, N, and P concentrations of *A. sparsifolia* in the study sites using the structural equation model. Significant regressions and nonsignificant regressions are shown by solid ($p < 0.05$) and dashed lines, respectively. The thickness of significant paths was scaled on the basis of magnitude of the standardized regression coefficient. Blue and red arrows denote positive and negative relationships, respectively. GD, groundwater depth; WPH, groundwater pH; WM, degree of mineralization of groundwater; SS, soil salt; SPH, soil pH; SC, soil C concentration; SN, soil N concentration; SP, soil P concentration; PC, plant C concentration; PN, plant N concentration; PP, plant P concentration; PNP, plant N:P ratio.

## 4. Discussion

### 4.1. Topsoil Nutrients Affected Variations in Leaf Ecological Stoichiometry

In contrast to our hypothesis, our results showed that soil P concentration had a direct negative effect on leaf N concentration in *A. sparsifolia* (Figures 3 and 5). In addition, the soil C concentration had a direct positive effect on the leaf C concentration of *A. sparsifolia* at a 1.3–2.2 m groundwater depth in the saturated aquifer zone (Figures 3 and 5). These results

agreed with previous experiments that showed that the soil P concentration was the vital factor influencing the growth and plant N and P concentrations of *A. sparsifolia* at different groundwater depths in a large area [5,8]. However, our results were not in agreement with the findings of Zhang et al. [5], who reported that soil P concentration was positively correlated to the leaf N concentration of *A. sparsifolia* and that the soil C concentration was not associated with the leaf C concentration of *A. sparsifolia* at a 2.5–11.0 m groundwater depth. One possible explanation for the different relationships was that topsoil nutrients had a greater effect on the leaf C and N concentrations of *A. sparsifolia* than groundwater in shallow groundwater depth environments. A previous study showed that soil mineralizes N leached into groundwater, which is the key source of nutrients for *A. sparsifolia* [13]. In the present study, *A. sparsifolia* was sampled at a 1.3–2.2 m groundwater depth in the saturated aquifer zone, which was shallower than that studied by Zhang et al. [5], i.e., a 2.5–11.0 m groundwater depth. In shallow groundwater environments, *A. sparsifolia* might absorb mineralized N in topsoil but not in groundwater due to the different nutrient uptake strategies in the saturated aquifer zone. This finding was consistent with the observation that plants uptake water and nutrients via the root system. The horizontal roots of *A. sparsifolia* quickly expand and tiller in shallow groundwater tables [30], and the vertical root depth of *A. sparsifolia* increases with increasing groundwater depth [30,31]. In addition, Gui et al. [15] reported that the soil water content in a shallow groundwater table (2.5 m) was higher than that in deep groundwater (4.5 and 11.0 m). In addition, our research team found that *A. sparsifolia* develops different efficient root architectures that absorb soil nutrients and water in response to nutrient and water availability, respectively [32]. Hence, *A. sparsifolia* has different nutrient uptake strategies to adapt to different groundwater depths and soil conditions in a desert ecosystem.

### 4.2. Groundwater Depth Influenced Variations in Leaf Ecological Stoichiometry

Previous studies showed that groundwater depth significantly influenced the leaf P concentration of *A. sparsifolia* but did not affect the leaf N concentration at 2.5–11.0 m [5]. Herein, we report that the leaf N concentration in *A. sparsifolia* was positively influenced by groundwater depth and that the leaf P concentration was not altered by groundwater depth in shallow groundwater habitats (1.3–2.2 m) (Figure 5). This finding was because *A. sparsifolia* absorbed more soil N in shallow groundwater environments than in deep groundwater. In a previous study, biological $N_2$ fixation accounts for more than 80% of the leaf N concentration in *A. sparsifolia* [12]. However, biological $N_2$ fixation requires more energy than nitrate uptake in natural ecosystems [33]. Hence, we believe that *A. sparsifolia* prefers to absorb N directly in soil for N fixation by the atmosphere in shallow groundwater environments. This finding was consistent with the observation that the topsoil around *A. sparsifolia* had a higher soil N concentration in the shallow groundwater table (2.5 m) than in deep groundwater (4.5 and 11.0 m) [34]. Moreover, the hydrochemical process of groundwater is significantly affected by topsoil nitrate and DOC concentrations [35]. Therefore, groundwater depth had different effects on the leaf N and P concentrations of *A. sparsifolia* in various habitats (shallow groundwater table or deep groundwater). These results also explained why topsoil nutrients were important for leaf carbon and nitrogen concentrations of *A. sparsifolia* in shallow groundwater depth environments. This adaptive mechanism could help phreatophytes to adapt to different hostile environments and explain their wide distribution in the extremely arid Taklimakan Desert.

### 4.3. Groundwater pH Affected Variations in Leaf Ecological Stoichiometry

In the Taklimakan Desert, surficial waters always have a high pH and are rich in sodium chloride and sulfate concentrations [36]. Thus, although plant roots have access to groundwater, the salinity of water affects plant growth [37]. In the present study, groundwater pH was positively related to the leaf C concentration of *A. sparsifolia* (Figures 3 and 5) and was significantly associated with the degree of mineralization of groundwater (Figure 3). This finding confirmed that the degree of mineralization of

groundwater is a vital factor that alters the growth of *A. sparsifolia* at shallow groundwater depths (1.3–2.2 m) in the Taklimakan Desert [38]. Herein, we found that groundwater characteristics affected the leaf nutrients of *A. sparsifolia*, but these effects were lower than those of soil nutrients. Further research is needed to elucidate the different nutrient uptake mechanisms of phreatophytes at various groundwater depths in arid desert ecosystems.

## 5. Conclusions

Topsoil nutrients drive leaf carbon and nitrogen concentrations in *A. sparsifolia* with shallow groundwater. In the saturated aquifer zone, topsoil nutrients have more influence on leaf nutrients of *A. sparsifolia* than groundwater depth, salinity, and soil salt concentrations. Groundwater depth and pH also affected the leaf carbon and nitrogen concentrations in *A. sparsifolia*. However, topsoil nutrients play a crucial role in influencing leaf nutrients in *A. sparsifolia* in shallow groundwater habitats. Phreatophytic *A. sparsifolia* preferred soil N absorption in shallow groundwater environments to groundwater and biological nitrogen fixation in deep groundwater. To our knowledge, our results provide a new adaptive strategy for nutrient utilization of a desert phreatophyte grown at shallow groundwater depths in the saturated aquifer zone in a desert–oasis ecotone. This study may contribute to the protection and restoration of phreatophytes in hyperarid desert ecosystems.

**Supplementary Materials:** The following are available online at https://www.mdpi.com/article/10.3390/w13213093/s1, Figure S1: Location of the study sites. MY, Moyu county; HT, Hetian county; CW, west of Cele county; CN, north of Cele county; CE, east of Cele county.

**Author Contributions:** Conceptualization, B.Z., X.G. and F.Z.; methodology, B.Z., X.G. and F.Z..; software, B.Z. and Z.Z.; investigation, B.Z., H.L., H.Y., Y.L., Z.Z. and J.X.; writing—original draft preparation, B.Z. and G.T.; writing—review and editing, C.H., Y.L., M.S., X.G. and F.Z.; All authors have read and agreed to the published version of the manuscript.

**Funding:** This study was funded by the Western Young Scholar Program-B of the Chinese Academy of Sciences (2018-XBQNXZ-B-018), the National Natural Science Foundation of China (31500367; 42071259), the Key Program of Joint Funds of the National Natural Science Foundation of China and the Government of Xinjiang Uygur Autonomous Region of China (U1603233), the Youth Innovation Promotion Association Foundation of the Chinese Academy of Sciences (2020435), the Third Batch of Tianshan Talents Program of Xinjiang Uygur Autonomous Region (2021–2023), and the Project for Cultivating High-Level Talent of Xinjiang Institute of Ecology and Geography, Chinese Academy of Sciences (E0502101).

**Institutional Review Board Statement:** Not applicable.

**Informed Consent Statement:** Not applicable.

**Data Availability Statement:** All data reported here is available from the authors upon request.

**Acknowledgments:** We thank Yonggang Li and Jiangshan Lai for assistance in data analysis and Mingfang Hu for assistance in soil and plant elemental analysis. We are also grateful to the anonymous referees for reviewing this manuscript.

**Conflicts of Interest:** The authors declare no conflict of interest.

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
