# Peer review of "Topsoil Nutrients Drive Leaf Carbon and Nitrogen Concentrations of a Desert Phreatophyte in Habitats with Different Shallow Groundwater Depths"

_water, doi:10.3390/w13213093_

Round 1

Reviewer 1 Report

Review

General comments

The research looks carefully conducted. Despite this, the potential paper shows problems on the description of groundwater/aquifer properties due to the expertise of the team on biology/ecology. Please, refer to my comments below to re-store the draft. The bibliography should be updated with recent literature and the language revised. After these implementations, the paper can be published in Water due to a good fit with the principal aim and objectives of the journal

Specific comments

Title

Lines 1-5. Re-ward the title avoiding this statement “Top soil nutrients are important”. This formulation sounds very subjective

Abstract

Lines 19-36. State if you refer to the groundwater in the saturated or un-saturated aquifer zone.

Lines 30-32. Revise the language. Pay attention to the repetition of the word “groundwater”.

  1. Introduction

Lines 72-74. Add these recent papers on variation of C, P and N concentrations in soil and groundwater with depths

- Holden, J., Grayson, R.P., Berdeni, D., Bird, S., Chapman, P.J., Edmondson, J.L., Firbank, L.G., Helgason, T., Hodson, M.E., Hunt, S.F.P. and Jones, D.T., 2019. The role of hedgerows in soil functioning within agricultural landscapes. Agriculture, Ecosystems & Environment, 273, pp.1-12.

- Medici, G., Baják, P., West, L.J., Chapman, P.J. and Banwart, S.A., 2021. DOC and nitrate fluxes from farmland; impact on a dolostone aquifer KCZ. Journal of Hydrology, 595, p.125658.

  1. Material and methods

Lines 113-128. State type of soil and bedrock aquifer (e.g., granite?)

Line 125. Depth interval 1.3 – 2.2 m. Are you studying the un-saturated zone of the aquifer? Are you describing perched aquifers above the water table?

  1. Results

Lines 171-256. Introduce more figures when you describe the results

Line 171. I prefer “soil and groundwater characteristics” mentioning geological bodies from the shallower to the deeper

Lines 172-182. Introduce ranges. Add pH average and ranges. pH acidic or basic?

Line 208. What do you mean by “significant” correlation? Please, refer to a statistical parameter

  1. Discussion

Line 262. Are you in the saturated or un-saturated zone? Are you dealing with a perched aquifer?

Lines 273-281. You mention groundwater at different depths. Are you dealing with the un-saturated or saturated aquifer zone? You have this problem throughout the manuscript

  1. Conclusion

Lines 323-332. You mention shallow and deep groundwater. Please, state if you’re dealing with the un-saturated or saturated portion of the aquifer

Line 344. Insert the recent publications on C, N and P in soil and groundwater that I have reported above

Figures and tables

Figure S1. Add the map of China with a dot on your study area

Reviewer 2 Report

The manuscript is well written and should be of intrest to your readers.

The manuscript would benifit from a read through for English usage.

The introduction is overly long and could be shortened considerably.

More detail in the experimental design, particularly replication number for each parameter tested is needed.

Round 2

Reviewer 1 Report

I have read again the manuscript, the authors managed to improve the manuscript. I have still four implementations

Abstract

"Degree of mineralization of groundwater; topsoil and pH". Un-clear what the authors mean by mineralization. Do you mean hydro-chemical compositions?

Introduction or Discussion

The authors do not mention that decreasing in P, N and C in groundwater with depth is a type common worldwide. Please, highlight this point, you can cite this paper and consider references there in. That's because the majority of the biomass is in the first cm of soil

Medici, G., Baják, P., West, L.J., Chapman, P.J. and Banwart, S.A., 2021. DOC and nitrate fluxes from farmland; impact on a dolostone aquifer KCZ. Journal of Hydrology595, p.125658

Conclusions

- This sentence is too long. The sentence needs to be divided in two parts and language must be improved

"Topsoil nutrients were more important for leaf carbon and nitrogen concentrations in A. sparsifolia than groundwater depth, salinity, and soil salt concentrations in environments with shallow groundwater in the saturated aquifer zone". 

Note that, your conclusion is very short and there is room for clarification

- Too many connectors in the conclusions (e.g., In addition, Moreover) close to each other

Reviewer 2 Report

N/A